# Layer-by-Layer Hollow Mesoporous Silica Nanoparticles with Tunable Degradation Profile

**DOI:** 10.3390/pharmaceutics15030832

**Published:** 2023-03-03

**Authors:** Jason William Grunberger, Hamidreza Ghandehari

**Affiliations:** 1Utah Center for Nanomedicine, Department of Molecular Pharmaceutics, University of Utah, Salt Lake City, UT 84112, USA; 2Department of Biomedical Engineering, University of Utah, Salt Lake City, UT 84112, USA

**Keywords:** silica nanoparticles, layer-by-layer, hollow, mesoporous, degradation, drug delivery

## Abstract

Silica nanoparticles (SNPs) have shown promise in biomedical applications such as drug delivery and imaging due to their versatile synthetic methods, tunable physicochemical properties, and ability to load both hydrophilic and hydrophobic cargo with high efficiency. To improve the utility of these nanostructures, there is a need to control the degradation profile relative to specific microenvironments. The design of such nanostructures for controlled combination drug delivery would benefit from minimizing degradation and cargo release in circulation while increasing intracellular biodegradation. Herein, we fabricated two types of layer-by-layer hollow mesoporous SNPs (HMSNPs) containing two and three layers with variations in disulfide precursor ratios. These disulfide bonds are redox-sensitive, resulting in a controllable degradation profile relative to the number of disulfide bonds present. Particles were characterized for morphology, size and size distribution, atomic composition, pore structure, and surface area. No difference was observed between in vitro cytotoxicity profiles of the fabricated nanoparticles at 24 h in the concentration range below 100 µg mL^−1^. The degradation profiles of particles were evaluated in simulated body fluid in the presence of glutathione. The results demonstrate that the composition and number of layers influence degradation rates, and particles containing a higher number of disulfide bridges were more responsive to enzymatic degradation. These results indicate the potential utility of layer-by-layer HMSNPs for delivery applications where tunable degradation is desired.

## 1. Introduction

Fabrication of layer-by-layer nanoparticles enables the assembly of multifunctional materials through the sequential deposition of thin films. The development of layer-by-layer nanoparticles allows for precise control of shell composition and thickness, and has been explored for applications in drug delivery, theranostics, and catalysis [1,2,3]. Advantages of layer-by-layer systems include protection of sensitive cargos such as RNA from nuclease degradation while in the bloodstream [4], controlled drug release through encapsulation of a drug-loaded core with a stimuli-responsive film [5], and optimization of target cell uptake by tuning the physicochemical properties of the outer layer such as charge and stiffness [6,7].

HMSNPs have shown significant potential in controlled drug delivery due to robust and low-cost synthesis scalable for industry with tunable physicochemical properties and high loading capacity. Silica-based nanocarriers have been designed to control drug release profiles through the use of stimuli-responsive gatekeepers [8]. Gatekeepers are added to the surface of mesoporous silica nanoparticles to physically block pores and prevent the release of cargo until exposed to specific stimuli, such as enzymes or pH, which results in the opening of the pores and subsequent drug release. However, these systems have limited ability for combination drug delivery due to complex synthetic methods and control of cargo compartmentalization. To improve the utility of HMSNPs, layer-by-layer systems can be used to overcome these limitations. Furthermore, the step-wise synthesis of layer-by-layer HMSNPs allows for the incorporation of different precursors and cargo within separate mesoporous shells for the development of highly tunable nanocarriers with the capacity to carry more than one drug.

Biological degradation and clearance of silica nanoparticles occur via hydrolysis of the siloxane framework into soluble silicic acid, which is nontoxic and excreted through the urine [9]. Long-term treatments that require frequent dosing regimens raise safety concerns regarding silica nanoparticle bioaccumulation [10]. To address this issue, the fabrication of nanoparticles with tunable degradation rates is preferable. Degradation kinetics are influenced by physicochemical properties of silica nanoparticles, such as size and porosity. One way to tune the degradation profile of silica nanoparticles is by incorporating redox-responsive disulfide-bridged silsesquioxanes [11,12,13]. Disulfide-bridged frameworks are cleaved by glutathione (GSH) in reductive intracellular environments. Redox-sensitive silica nanoparticles have been investigated for controlled drug delivery to reduce premature release in the bloodstream until cellular uptake occurs where high GSH concentrations, up to 10 mM, trigger cargo release [14]. While many redox-responsive silica nanoparticles have been developed, they mostly rely on a “gatekeeper” mechanism for drug release [15]. By using several layers of redox-responsive silica shells, the use of gatekeepers can be avoided, along with their inherent limitations.

To develop robust layer-by-layer HMSNPs for drug delivery applications, it is essential to examine the particles’ degradation profile. Herein, two types of redox-sensitive layer-by-layer HMSNPs were designed to investigate the effects of particle composition and the number of layers on degradation profiles. Establishing the synthesis and characterization of layer-by-layer silica nanoparticles and their degradation rates paves the way for combination drug delivery and predetermined controlled release from such systems.

In this article, we report the synthesis and characterization of two types of layer-by-layer HMSNPs by varying the silane precursor ratios of tetraethyl orthosilicate and bis[3-(triethoxysilyl)propyl] disulfide. Particles were characterized by dynamic light scattering, scanning transmission electron microscopy, energy dispersive X-ray spectrometry, thermogravimetric analysis, powder X-ray diffraction, and nitrogen adsorption-desorption for investigation of particle morphology, atomic composition, pore structure, and surface area. The degradation profiles of the fabricated particles were investigated in simulated body fluid (SBF) with 10 mM GSH and their cell toxicity in macrophages was further evaluated. We observed distinct degradation profiles for all three fabricated nanoparticles. The addition of subsequent layers increased particle stability and a significant increase in degradation rate in the presence of glutathione.

## 2. Materials and Methods

### 2.1. Materials

The following materials were purchased from Fisher Scientific (Pittsburgh, PA, USA): hydrochloric acid (HCl), ammonium hydroxide, sodium hydroxide (NaOH), glutathione (GSH), RPMI-1640 medium, fetal bovine serum (FBS), and TrypLE. Sodium carbonate was purchased from EMD Millipore (Billerica, MA, USA). Bis[3-(triethoxysilyl)propyl] disulfide (BTEPDS) was obtained from Gelest (Morrisville, PA, USA). Tetraethyl orthosilicate (TEOS), triethylamine (TEA), and hexadecyltrimethylammonium bromide (CTAB) were obtained from Sigma-Aldrich (St. Louis, MO, USA). Ethanol 95% and absolute ethanol were purchased from Decon Labs (King of Prussia, PA, USA). RAW 264.7 macrophage cell line was received from ATCC (Manassas, VA, USA). The cell counting kit-8 (CCK-8) cytotoxicity assay was purchased from Dojindo (Rockville, MD, USA). Simulated body fluid (SBF) pH 7.4 was purchased from Biochemazone (Waterloo, ON, Canada). SBF was used due to the ionic strength and composition closely resembling physiological conditions compared to other aqueous mediums such as phosphate buffer saline. The pH of SBF with 10 mM GSH was adjusted to pH 7.4 with NaOH.

### 2.2. Methods

#### 2.2.1. Synthesis of Layer-by-Layer HMSNPs

The process for the synthesis of layer-by-layer HMSNPs is depicted in Figure 1 and described below:Stöber nanospheres were prepared as follows: 3.6 mL ammonium hydroxide, 2.8 mL deionized water, and 100 mL absolute ethanol were mixed in a 250 mL Erlenmeyer flask for 10 min at a stirring rate of 400 rpm. Next, 3.5 mL TEOS was added dropwise and the reaction was left stirring for 24 h at room temperature. The synthesized particles were collected by centrifugation using an Avanti J-15R (Beckman Coulter Inc., Indianapolis, IN, USA) at 11,420 RCF for 10 min, and washed thrice with deionized water and 95% ethanol. Final particles were suspended in 50 mL deionized water.Mesoporous coated Stöber SNPs were prepared as follows: 2.73 mL absolute ethanol, 20 mL deionized water, 27 μL TEA, and 70 mg CTAB were mixed in a 100 mL round bottom flask under a stirring rate of 600 rpm in an oil bath with reaction temperature at 80 °C, using a water condenser, for 30 min. 10 mL of previously synthesized Stöber SNPs were added to the solution and left stirring for 15 min. Stirring rate was increased to 1400 rpm and 87.5 μL TEOS and 37.5 μL BTEPDS were added simultaneously to the solution. The reaction was left stirring for 3 h. The synthesized particles were collected by centrifugation, washed once with 95% ethanol, and suspended in 10mL deionized water.HMSNPs (H_1_) were prepared as follows: 1540 mg sodium carbonate was dissolved in 10 mL deionized water in a 100 mL round bottom flask under a stirring rate of 1200 rpm with reaction temperature at 50 °C. After 1 h, the previously fabricated mesoporous coated Stöber SNPs were added and left under stirring for 10 h. The particles were collected by centrifugation and washed thrice with deionized water and 95% ethanol. To remove the CTAB, particles were suspended in acidic ethanol (1 mL concentrated HCl in 30 mL absolute ethanol) and refluxed at 80 °C for 6 h. This CTAB removal process was performed twice. Particles were washed thrice with deionized water and 95% ethanol. Final particles were suspended in 10 mL deionized water.Double-layered HMSNPs (H_2_) were fabricated as follows: 2.73 mL absolute ethanol, 20 mL deionized water, 27.5 μL TEA, and 70 mg CTAB were mixed in a 100 mL round bottom flask under a stirring rate of 600 rpm in oil bath with reaction temperature at 80 °C, using a water condenser, for 30 min. The previously synthesized H_1_ was added to the solution and left stirring for 15 min. Stirring rate was increased to 1400 rpm and 142 μL BTEPDS was added to the solution. The reaction was left stirring for 3 h. The synthesized particles were collected by centrifugation, washed thrice with water and 95% ethanol, and CTAB was removed via acidic ethanol wash twice. Particles were washed thrice with deionized water and 95% ethanol. Final particles were suspended in 10 mL deionized water.Triple-layered HMSNPs (H_3_) were fabricated as follows: 2.73 mL absolute ethanol, 20 mL deionized water, 27.5 μL TEA, and 70 mg CTAB were mixed in a 100 mL round bottom flask under a stirring rate of 600 rpm in oil bath with reaction temperature at 80 °C, using a water condenser, for 30 min. The previously synthesized H_2_ was added to the solution and left stirring for 15 min. Stirring rate was increased to 1400 rpm and 140 μL TEOS and 60 μL BTEPDS were added simultaneously to the solution. The reaction was left stirring for 3 h. The synthesized particles were collected by centrifugation, washed thrice with water and 95% ethanol, and CTAB was removed via acidic ethanol twice. Particles were washed thrice with deionized water and 95% ethanol. Final particles were suspended in 10 mL deionized water.

**Scheme 1 pharmaceutics-15-00832-sch001:**
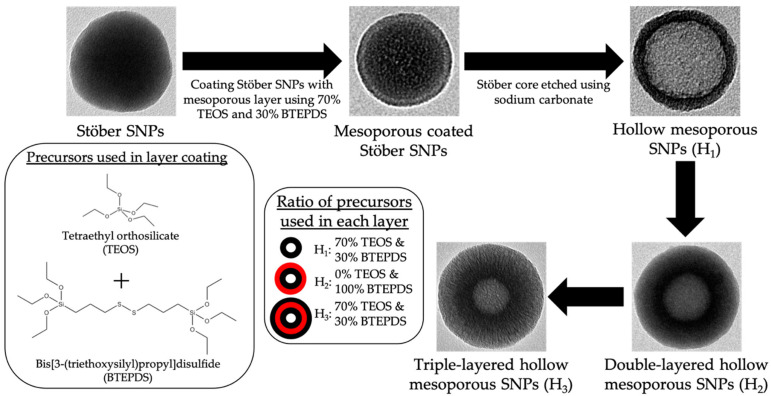
Top panel: the process of fabricating HMSNPs (H_1_) starting from a dense Stöber core. Bottom panel from left to right: the structure of the alkoxysilane precursors TEOS and BTEPDS, precursor ratios for outer shell layer of each nanoparticle with black circles representing 70% TEOS & 30% BTEPDS and red circles representing 100% BTEPDS, differentiating HMSNPs into layer-by-layer HMSNPs: H_2_ as double-layered particles; and H_3_ as triple-layered particles.

#### 2.2.2. Physicochemical Characterization of Nanoparticles

A JEOL JEM-1400 Plus Transmission Electron Microscope (Peabody, MA, USA) was used for acquiring transmission electron microscopy (TEM) images. A 200 kV JEOL JEM-2800 Scanning/Transmission Electron Microscope combined with a dual detector Energy Dispersive Spectrometer (EDS) was used to obtain scanning transmission electron microscopy (STEM) images and EDS spectra of the fabricated nanoparticles. Samples were suspended in 95% ethanol and drop-casted on TEM grids (Ted Pella Inc., prod. number 01824, Redding, CA, USA) for all electron microscopy analysis. EDS spectral images were processed as net-counts (background subtracted) maps with a 7 by 7 kernel size, 10 sec frame time, with the number of frames ranging from 8 to 20, and 256 × 256 pixels. For evaluating pore size and surface area, a Micromeritics 3Flex Sorption Analyzer (Norcross, GA, USA) was used to perform nitrogen adsorption−desorption isotherms. Before analysis, the samples were dried completely by bubbling dry N_2_ at 90 °C. The specific surface areas were calculated using the Brunauer−Emmett−Teller (BET) method. Utilizing the Barrett-Joyner-Halenda (BJH) technique, the pore size and pore volume of the synthesized nanoparticles were obtained from the adsorption branch of the isotherm. Thermogravimetric Analysis (TGA) was performed using TA Instruments Discovery SDT 650 (New Castle, DE, USA). TGA measurements were studied under an N_2_ atmosphere using a temperature range of 30 to 800 °C with a ramp of 20 °C min^−1^. Using Dynamic Light Scattering (DLS) in a Malvern Zetasizer Nano ZS (Worcestershire, UK), the zeta potential and hydrodynamic diameter of the fabricated nanoparticles were measured in 10 mM NaCl and deionized water respectively. A Bruker D2 Phaser (Madison, WI, USA) was utilized to investigate X-ray Diffraction (XRD) patterns of the synthesized particles using Cu Kα radiation (λ = 0.1542 nm) at 30 kV and 10 mA. The XRD spectra were recorded at a scanning speed of 0.01°/step, with a step size of 0.02°/step in a 2 theta scattering angle ranging from 2 to 8.

#### 2.2.3. Cytotoxicity Assay

Cytotoxicity of the fabricated nanoparticles was evaluated on RAW 264.7 murine macrophages. Macrophages were cultured in RPMI-1640 medium containing 10% FBS and passaged every 2–3 days using TrypLE to detach the cells. Cells were counted via trypan blue staining and read by Invitrogen Countess 3 Automated Cell Counter (Thermo Fisher Scientific, Waltham, MA, USA). Macrophages were seeded into 96-well plates at a density of 10,000 cells per well and incubated for 24 h at 37 °C in 5% CO_2_. The media was aspirated and the nanoparticles were suspended in complete media with concentrations ranging from 5 to 500 μg mL^−1^ in each well and incubated for 24 h. Cell viability was determined by a CCK-8 assay. Experiments were conducted in triplicate.

#### 2.2.4. Evaluation of In Vitro Degradation Profiles

Degradation profiles of all three fabricated HMSNPs (H_1_, H_2_, H_3_) were evaluated by suspending 10 mL SBF pH 7.4 with GSH concentrations mimicking intracellular and extracellular reductive environments (10 mM and 0 mM, respectively) at a concentration of 100 μg mL^−1^ in HDPE scintillation vials and incubated under constant rotation of 150 rpm at 37 °C. At time points 0, 2, 4, 6, 12, 24, 48, 72, 168, and 336 h, 1.5 mL of sample was added to polypropylene tubes and centrifuged at 1600 RCF for 20 min. The top 1 mL of supernatant was collected for silicon (Si) measurement using inductively coupled plasma mass spectrometry (ICP-MS). The remaining 0.5 mL of milieu and pellet was resuspended in 1 mL fresh SBF or SBF with 10 mM GSH, to maintain sink conditions and replenish reduced GSH, and added back into the scintillation vials. Media samples were diluted by a factor of 5 as follows: 1 mL of media was transferred into a polystyrene tube, to which 200 μL of concentrated HNO_3_ and 100 μL of concentrated HF were mixed in, and then diluted to 5 mL with 2.4% HNO_3_. Determination of Si was performed with a triple quadrupole inductively coupled plasma mass spectrometer (ICP-MS, Agilent 8900, Santa Clara, CA, USA) at the ICP-MS labs, Dept. of Geology and Geophysics, University of Utah. An external calibration curve was prepared from 1000 mg L^−1^ Si standard (Inorganic Ventures, Christiansburg, VA, USA), using the same HNO_3_ plus HF matrix used for media. Si standard was added at 10 ng mL^−1^ to media samples, calibration solutions, and blanks as an internal standard and analyzed in the ICP-MS using a cyclonic PTFE spray chamber, PTFE nebulizer and dual-syringe introduction system (Teledyne, AVX 71000, Thousand Oaks, CA, USA), platinum cones and sapphire injector in a platinum-shielded quartz torch. Mass 28 and a flow of 2 mL H_2_/min in the collision cell were used to monitor Si intensity. The limit of detection (LoD) was calculated as three times the standard deviation of the blanks, multiplied by the dilution factor used for samples (5). The instrument is located in a filtered air positive pressure lab and sample handling and dilutions were performed in laminar flow benches using calibrated pipettors (Eppendorf Reference, Hamburg, Germany). All chemicals were trace metal grade purity. Experiments were conducted in triplicate.

#### 2.2.5. Statistical Analysis

GraphPad Prism 9 software (San Diego, CA, USA) was used to determine statistical significance. Results obtained are expressed as mean ± standard deviations (SD). The difference between groups was determined by analysis of variance (ANOVA), followed by Tukey’s multiple comparisons test. Differences with *p* value < 0.05 were considered as significant.

## 3. Results and Discussion

### 3.1. Synthesis and Characterization of Layer-by-Layer HMSNPs

The nanoparticles designed in this study were prepared in a five-step process, resulting in the fabrication of single-layered (H_1_), double-layered (H_2_), and triple-layered (H_3_) hollow mesoporous silica nanoparticles as shown in Figure 1. All mesoporous layers were synthesized using the modified Stöber method through base-catalyzed condensation of the silane precursors in the presence of TEA and water in ethanol using the surfactant CTAB as the structure-directing agent. The silane precursors utilized were TEOS and BTEPDS. The average diameter of these layer-by-layer HMSNPs ranged from approximately 110 to 170 nm, as each subsequent layer deposition increased particle size. Fabrication of triple-layered HMSNPs was achieved through subsequent mesoporous layer deposition resulting in the final particle (H_3_) with an inner layer of 70% TEOS/30% BTEPDS, a middle layer of 100% BTEPDS, and an outer layer of 70% TEOS/30% BTEPDS.

The morphology of layer-by-layer HMSNPs was analyzed using electron microscopy (Figure 1). TEM images of all three nanoparticles show uniform hollow spherical morphologies with increasing size and shell thickness as the number of layers increase. Electron microscopy images were processed with Fiji software to measure particles’ diameters and shell thickness as shown in Table 1. The average diameters and shell thicknesses of H_1_, H_2_, and H_3_ were 106 ± 12 nm and 10 ± 1 nm, 131 ± 14 nm and 33 ± 4 nm, and 171 ± 8 nm and 55 ± 2 nm, respectively. The darkfield and secondary electron STEM images show a porous shell with disordered pore structure and increasing particle density as the number of layers increase.

DLS was used to determine zeta potential, the hydrodynamic diameter and polydispersity by the number-weighted distribution (Table 1). H_1_, H_2_, and H_3_ particles had average hydrodynamic diameters in deionized water of 132 ± 41 nm, 196 ± 56 nm, and 224 ± 76 nm, respectively. Zeta potentials were measured in 10 mM NaCl (pH 7.2) at 25 °C as described in PCC-2 of the Nanotechnology Characterization Laboratory [16]. The zeta potentials of H_1_, H_2_, and H_3_ were −16 ± 3 mV, −30 ± 4 mV, and −22 ± 3 mV, respectively. As particle disulfide concentration increases, the zeta potential becomes more negative, which is consistent with previous observations potentially due to changes in surface silanol density [17].

EDS was used for evaluating particle compositions by measuring atomic densities. The EDS spectra demonstrate the homogenous distribution of sulfur throughout each layer, with distinct boundaries of all layers visible, indicating the successful fabrication of our layer-by-layer HMSNPs through sequential layer deposition (Figure 2). The increased atomic density of sulfur within the second layer compared to the first and third is to be expected as this layer is fabricated with a higher overall ratio of the disulfide precursor BTEPDS.

To process the relative composition of silicon to sulfur, EDS images of whole particles were processed as net counts to measure percent atomic concentrations (Table 2). The total sulfur composition of H_2_ was significantly higher than H_1_ and H_3_ due to the increase in BTEPDS used for the fabrication of the second layer. While the outer layers of H_1_ and H_3_ contain the same sulfur composition, the atomic sulfur percentage of H_3_ is also significantly higher compared to H_1_, which is to be expected due to this second layer.

Nitrogen adsorption-desorption isotherm analysis was used to investigate the specific surface area and pore size distribution of the fabricated nanoparticles (Table 3). The average pore diameter of H_1_ was 5.7 nm, while H_2_ and H_3_ were observed to have narrower mesopores of 3.9 nm and 3.6 nm, respectively. The nitrogen adsorption-desorption isotherms and pore size distributions are illustrated in Figure 3, panels A, B, and C. According to The International Union of Pure and Applied Chemistry (IUPAC) classification system for adsorption isotherms [18], H_1_ exhibits type IV isotherms, while H_2_ and H_3_ exhibit type I isotherms. Both isotherm types are characteristic of mesoporous materials, but the collapse of the hysteresis loop observed in the isotherms of H_2_ and H_3_ is indicated by pronounced narrowing of pore diameter, evident in the BJH pore size (Table 3). XRD patterns shown in Figure 3D lack the presence of Bragg peaks for all three particles at low-angle measurements, indicative of a disordered pore structure. This is commonly observed in organosilica nanoparticles with high concentrations of BTEPDS due to the bulky disulfide groups disrupting mesostructure ordering [19]. TGA of the particles is plotted in Figure 3E. The graphs indicate the weight loss of each nanoparticle between 30 °C to 800 °C. The most significant weight loss (ca. 49%) was observed for H_2_. This is due to the high content of organosilyl chains (-Si-C-C-C-S-S-C-C-C-Si-) embedded in the framework of the fabricated nanoparticles. As expected, the smallest weight loss (ca. 7%) was observed for H_1_, while the weight loss (ca. 30%) observed for H_3_ fell between the other two nanoparticles.

### 3.2. In Vitro Cytotoxicity

For evaluating the cytotoxicity of the layer-by-layer HMSNPs, RAW 264.7 macrophages were utilized. H_1_, H_2_, and H_3_ nanoparticles were suspended in complete media and added to the cells in the concentration range of 5 µg mL^−^^1^ to 500 µg mL^−^^1^ (Figure 4). Relative cell viability was analyzed via CCK-8 assay. No toxicity was observed in concentrations below 100 µg mL^−^^1^ for H_1_, while toxicity was observed at and above 100 µg mL^−^^1^ for particles H_2_ and H_3_ with significant differences compared to H_1_.

### 3.3. Evaluation of In Vitro Degradation Profiles

To determine the degradation profiles of H_1_, H_2_, and H_3_ in redox conditions mimicking intracellular and extracellular environments, free silicon was quantified by ICP-MS over 14 days and is shown in Figure 5. “Passive” dissolution of silica nanoparticles in aqueous media is a three-step process. Firstly, water molecules adsorb onto the siloxane framework. Secondly, hydrolysis of siloxane into silanols occurs. Thirdly, ion-exchange through nucleophilic attack of another hydroxyl group results in the formation of silicic acid which is free to leach out to the medium in the form of Si(OH)_4_ [20]. As shown in Figure 5, H_1_ nanoparticles degrade rapidly in both SBF and SBF with 10 mM GSH (ca. 25%) within the first 48 h. This rapid degradation is a result of the “passive” dissolution of these nanoparticles due to a more accessible surface area for water molecules to absorb to compared to H_2_ and H_3_ nanoparticles. H_1_ nanoparticles were observed to rapidly collapse in both degradation media (Figure 6A), suggesting that degradation is predominantly due to hydrolysis. The change in degradation profiles of H_2_ is most significant in the presence of GSH, presumably due to redox-triggered biodegradation of the disulfide-bridged framework which increased because more BTEPDS was used for fabrication of H_2_, whereas the rate of hydrolytic degradation is reduced due to the hydrophobic nature of the disulfide-bridged siloxane. Further evidence that the observed differences of degradation profiles in SBF alone are a result of “passive” dissolution is the correlation between particles’ BET surface area and degradation, as surface area decreases so does degradation. All three particles exhibited significant increases in total degradation after 14 days in the presence of GSH compared to SBF alone. Further evidence of increased degradation in the presence of GSH is observed through TEM images of H_2_ and H_3_ nanoparticles (Figure 6B,C, respectively), in which particles begin to visibly degrade and break by day 7, and by day 14 significant silica debris are present. While most particles are intact after 14 days in both SBF and SBF with 10 mM GSH, more degradation occurs in the presence of GSH. The higher magnification in Figure 6C for day 7 and day 14 in SBF alone exhibits an increase in surface roughness, representative of slow surface erosion.

The above results demonstrate that redox-triggered degradation profiles of HMSNPs can be altered by changing the ratio of sulfur to silicon and the number of layers during the nanoparticle fabrication process. Ultimately, understanding how to alter biodegradation rates through facile synthetic methods would help us design the right nanoparticle for utility in specific applications. For example, they can be used in predetermined combination drug delivery by incorporating different bioactive agents in different layers. To accomplish this, the outer layer must degrade faster and release the first cargo, while the inner layers degrade second to release the second cargo, as evident in H_3_ nanoparticles illustrated in Figure 2.

Temporal control in combination delivery of chemotherapeutics for example is one promising application for these systems as previous studies that have used sequential application of chemotherapeutics have shown increased efficacy [21,22,23]. Other applications of layer-by-layer silica nanoparticles would be in predetermined controlled release and chronotherapy where different rates of release at different times of one or a combination of drugs are desired [24]. Examples of traditional layered dosage forms designed for predetermined drug release include methylphenidate extended-release tablets (Concerta^®^ XL) where an initial bolus release is followed by constant release using an osmotic pump. Nanoparticles have the additional advantage of controlling the location of drug release by tuning their size as well as surface properties [25,26]. Our layer-by-layer system could be designed to achieve spatio-temporal release by tailoring the degradation rates of each layer through adjusting disulfide composition, as well as the size and surface properties of the particles to control biodistribution and accumulation at target sites.

While these systems are promising, future studies are needed to further elucidate changes in degradation profiles proportional to changes in TEOS to BTEPDS ratios during synthesis. Since the double and triple layered nanoparticles were still intact after 14 days, longer in vitro studies are necessary to observe complete degradation. Most importantly, future in vivo nanoparticle degradation analysis is crucial to understand the biological fate of the degradation products and the effect of composition on the release of bioactive agents from the nanoparticles.

## 4. Conclusions

We have developed two types of layer-by-layer HMSNPs exhibiting tunable degradation. In summary, we fabricated two novel HMSNPs, characterized their physicochemical properties, investigated in vitro cytotoxicity, and determined their degradation profiles in conditions mimicking intracellular and extracellular reducing environments. Toxicity in RAW 264.7 macrophages was not observed for all nanoparticles after incubation over 24 h in nanoparticle concentration ranges of 5 to 100 µg mL^−^^1^. Results from the degradation study indicate that by altering the disulfide composition and number of layers, it is possible to manipulate the degradation profile of the particles. Increasing the disulfide concentration reduced hydrolytic degradation attributed to the hydrophobicity of disulfide-bridged silsesquioxane framework, but redox-triggered biodegradation was enhanced significantly. Lastly, as the number of mesoporous silica layers increased, degradation decreased presumably due to a reduction in solvent accessibility of the siloxane framework, enhancing particle stability. Further investigations elucidating percent disulfide composition on degradation rate are needed to provide key information for designing tailored redox-sensitive layer-by-layer HMNSPs for predetermined spatio-temporal release of bioactive agents.

## Data Availability

The raw data presented in this study are available from the corresponding author on request.

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
