# Peer review of "Layer-by-Layer Hollow Mesoporous Silica Nanoparticles with Tunable Degradation Profile"

_pharmaceutics, 2023, doi:10.3390/pharmaceutics15030832_

Round 1
Reviewer 1 Report
The manuscript describes the synthesis and properties of three hollow mesoporous silica nanoparticles built layer-by-layer with one, two or three layers. Although the work is interesting and well conducted some minor revision is still required before acceptance for publication:
1)Lines 15-17; clarify in the abstract the role of disulfide precursors;
2) A scheme should be inserted in the Introduction clarifying chemical structures and reactions responsible for the synthesis;
3)After line 73, please, give a brief overview of the main results achieved in this work.
4)Line 233: please, insert "microscope"after "Electron..."
5)Lines 340-348: In Figure 5 please, increase sizes for nimbers and meaning for the symbols in the figure to improve their readability.
6)For Figure captions in general: please, always provide meaning for all abbreviations used for the sake of readability and clarity.
7)Line 391: Insert "nanoparticles"before "concentrations..."
Reviewer 2 Report
The manuscript entitled “Layer-by-Layer Hollow Mesoporous Silica Nanoparticles with Tunable Degradation Profile” approaches the exciting field of silica nanomaterials. The authors studied the degradation profile of several silica “coatings” by tuning the shell thickness and its composition by incorporating redox-responsive organoalkoxysilanes. The topic is timely, but in my opinion is not suitable for publication in Pharmaceutics because it is focused on the development of the material and drug release studies are missing in the manuscript. The manuscript is more interesting for the readers of a nanomaterials journal. Nevertheless, I list my specific concerns for the manuscript below. My comments are mainly aimed at tightening up the logic and clarity of what you have communicated:
Major Comments:
1. A couple of articles are missing in the introduction section. For example, Luisa de Cola and others, developed several approaches to prepare silica nanoparticles with enhanced degradation in vitro, by incorporation into the silica framework alkoxysilicate precursors containing for example disulfide bridges [1,2], enzymatically degradable bonds, pH-cleavable linkers and light responsive moieties.
[1] DOI: https://doi.org/10.1039/C5NR09112H
[2] DOI: https://doi.org/10.1021/acsami.7b04351
2. Gao et al, studied the degradation of silica nanoparticles by incorporating in the silica matrix carbamate bonds (https://doi.org/10.1039/C7RA12377A). These particles took 21 to 90 days to degrade completely, and the degradation rate increased by changing the pH. The authors should have studied the degradation rates over more days.
3. There is a review article from 2017 that reviewed redox-responsive MSNs, that is missing in the reference section. https://doi.org/10.3390/nano11092222
4. On the introduction section (Line 40), the authors wrote that “Silica-based nanocarriers…the use of stimuli-responsive gatekeepers”. But we can see in my previous comments that there are more than gatekeepers to control the drug release on silica nanoparticles. The silica matrix can also be stimuli responsive. In fact, this is what the authors are doing, a stimuli-responsive shell.
5. It should also be interesting to compare the responsive silica shell with a non-responsive one, for example, a silica shell with the same thickness but only with TEOS as the silica source.
6. Scheme 2 for example is the type of scheme that works well in a project proposal, to combine the previous results with future work. But this is what this manuscript is missing, the release of a drug or a dye, or two different dyes for example one for each layer. In the following paper the authors also tune the shell thickness of hollow mesoporous silica nanoparticles and use DOX to study the loading and releasing behavior. DOI: https://doi.org/10.1021/acsami.6b13876
7. In the materials section is more usual to wrote absolute ethanol than “ethanol 200 proof”.
8. In the materials section the authors should write the mass or mL rather than the amount in mmol. It would be easier to reproduce and less prone to mistakes if someone wants to use this methodology.
9. Why the authors suspended the nanoparticles in deionized water?
10. On page 3, line 103. Do we really need the 80C? Have the authors tried other temperatures? From my experience, the problems in reproducibility of silica-based nanoparticles are related to the temperature. Everyone reports the 80C, but the synthesis usually occurs at lower temperatures because the authors are reporting the temperature of the oil bath and not the temperature inside the reaction flask.
11. On page 3, line 108. Careful that ethanol is known to remove CTAB from the silica pores.
12. Scheme 1 could be improved. As this is just a scheme the authors could combine the TEM images with a scheme (such as the one for H1, H2 and H3 but higher in size and next to the TEM images).
13. Why H3 has a more positive zeta potential than H2? Because H3 is composed mainly of TEOS and H2 is composed mainly of the redox-responsive organoalkoxysilane. I would aspect that the higher the organic content, more positive the zeta-potential.
14. Page 9, Line 324. H1 nanoparticles degrade rapidly in both SBF and SBF with GSH. What about nanoparticles with only TEOS as the silica source? Could they also suffer from passive dissolution due to a thin shell?
Reviewer 3 Report
The article entitled "Layer-by-Layer Hollow Mesoporous Silica Nanoparticles with Tunable Degradation Profile" is about an interesting and promising scenario with the use of silica nanoparticles designed to chemically response and be exploitable for drug delivery in biomedical applications. This reviewer thinks about the article is successful in describing and spreading the scientific research with high quality. Some corrections have been underlined by this referee:
Abstract: Correct designing with "the design of such nanostructures for controlled combination drug delivery would benefit from minimizing degradation and cargo release in circulation while increasing intracellular biodegradation” Introduction: lines 29-37 The authors introduce the use of “nanoparticles” of different nature, not necessarily silica-based, their general use and advantages for the stated applications. It seems a bit too generalist, some insight into WHICH properties are related to WHAT useful features of the nanoparticles would help the reader to understand the challenges tackled in this work. lines 38-47 Again, some insight about the limitations of current silica-based delivery systems would be of use, to better understand the advantages that layer-by-layer synthesis brings to the table. line 41: Maybe explain the concept of “stimuli-responsive gatekeeper”, to broaden the reachable audience of the article. Please revise lines 51-53: “Physicochemical properties of silica nanoparticles influence degradation kinetics. As such it is preferable to design delivery systems with physicochemical properties that tune the degradation rate to avoid concerns of silica nanoparticle bioaccumulation [10].”
Materials and Methods, line 76: Suggestion: “The following materials were purchased (not obtained) from Fisher Scientific…”. line 95: Quantities of solvent (deionized water, ethanol, etc), when not taking part as reagents are usually indicated in volume, instead of mmol, for practical reasons. Same for lines 101, 102, 118, 119, 127 and 128. Line 114 Please specify, if relevant, the concentration of HCl used. Lines 157-158 Were the DLS and Zeta Potential measurements performed in deionized water? Maybe specify.
Results and Discussion; Table 1: There seems to be some format problems with this table. Also about the size data from DLS measurements, is it from intensity, volume, number? Maybe be precise in reporting this information. It seems unexpected that the 2-layer particles (H2) exhibit the higher value of Zeta-Potential, thus being more stable in solution, given they are the ones to exhibit a higher amount of hydrophobic groups to the medium (disulphide bonds). Some discussion might be necessary. Figure 5 The legend is too small. Scheme 2 Caption “…and release the other cargo (shown in green dots) slower more slowly”
General Comment: it would be interesting to try out different concentrations of GSH to better understand the sensibility of the system to the redox degradation, or else, to put the system in contact with a cell lysate, as a trial to better reproduce the conditions inside a real cell.

Round 2
Reviewer 2 Report
The authors have satisfactorily addressed most of my concerns, although the manuscript is more or less the same. I think that a control experiment with particles built only with TEOS as the silica source is still missing. I still think that scheme 2 is not necessary and I have a few minor questions about Scheme 1, Figure 5 and Figure 6.
- Scheme 1 is not clear. I will suggest removing the structure of TEOS and BTEPDS, or at least remove the TEOS structure. At 100% zoom, we can barely read the layers composition. It is not clear if H2, has only a layer with 0%TEOS and 100% BTEPDS, or two layers, the first with 70% TEOS and 30% BTEPDS and the second one with 0% TEOS and 100% BTEPDS. The scheme is two small, and it is not clear. Maybe use different colors for each layer and a thick line would help. Usually, people before reading a manuscript they look to the figures.
- In Figure 5, the authors are fitting an equation to the data or just connecting the dots? Connecting the dots can lead to misleading results since we do not know what happen in the meantime.
- In figure 6, H1 (Fig. 6 A) the particles do not look the same as in Figure 1A, mainly because in Figure 1 we see only one particle. In addition, for example in Figure 6C, the particles look more degraded on day 7 than on day 14. Again, if the images are only showing a couple of particles, or one, it is hard to have a sense of the degradation behavior with time. I have seen TEM images of silica nanoparticles that degrade over time, and usually the images start to have a lot of silica debris.
